# Model Comparison of Heritability Enrichment Analysis in Livestock Population

**DOI:** 10.3390/genes13091644

**Published:** 2022-09-13

**Authors:** Xiaodian Cai, Jinyan Teng, Duanyang Ren, Hao Zhang, Jiaqi Li, Zhe Zhang

**Affiliations:** Guangdong Provincial Key Laboratory of Agro-Animal Genomics and Molecular Breeding, College of Animal Science, South China Agricultural University, Guangzhou 510642, China

**Keywords:** heritability enrichment, genetic architecture, LD, livestock, complex trait

## Abstract

Heritability enrichment analysis is an important means of exploring the genetic architecture of complex traits in human genetics. Heritability enrichment is typically defined as the proportion of an SNP subset explained heritability, divided by the proportion of SNPs. Heritability enrichment enables better study of underlying complex traits, such as functional variant/gene subsets, biological networks and metabolic pathways detected through integrating explosively increased omics data. This would be beneficial for genomic prediction of disease risk in humans and genetic values estimation of important economical traits in livestock and plant species. However, in livestock, factors affecting the heritability enrichment estimation of complex traits have not been examined. Previous studies on humans reported that the frequencies, effect sizes, and levels of linkage disequilibrium (LD) of underlying causal variants (CVs) would affect the heritability enrichment estimation. Therefore, the distribution of heritability across the genome should be fully considered to obtain the unbiased estimation of heritability enrichment. To explore the performance of different heritability enrichment models in livestock populations, we used the VanRaden, GCTA and α models, assuming different α values, and the LDAK model, considering LD weight. We simulated three types of phenotypes, with CVs from various minor allele frequency (MAF) ranges: genome-wide (0.005 ≤ MAF ≤ 0.5), common (0.05 ≤ MAF ≤ 0.5), and uncommon (0.01 ≤ MAF < 0.05). The performances of the models with two different subsets (one of which contained known CVs and the other consisting of randomly selected markers) were compared to verify the accuracy of heritability enrichment estimation of functional variant sets. Our results showed that models with known CV subsets provided more robust enrichment estimation. Models with different α values tended to provide stable and accurate estimates for common and genome-wide CVs (relative deviation 0.5–2.2%), while tending to underestimate the enrichment of uncommon CVs. As the α value increased, enrichments from 15.73% higher than true value (i.e., 3.00) to 48.93% lower than true value for uncommon CVs were observed. In addition, the long-range LD windows (e.g., 5000 kb) led to large bias of the enrichment estimations for both common and uncommon CVs. Overall, heritability enrichment estimations were sensitive for the α value assumption and LD weight consideration of different models. Accuracy would be greatly improved by using a suitable model. This study would be helpful in understanding the genetic architecture of complex traits and provides a reference for genetic analysis in the livestock population.

## 1. Introduction

Heritability enrichment analysis is a widely used method to explore the genetic architecture of human complex traits [1,2,3], and it is typically defined as the proportion of variability in heritability of the category subset divided by the proportion of single nucleotide polymorphism (SNP). Heritability enrichment reflects the relative importance of a specific category of functional genomic regions or genetic variates in terms of heritability. Over the past few years, explosively increased omics data have discovered many functional variant/gene subsets, biological networks and metabolic pathways which likely affect complex traits or diseases. For example, in human ENCODE, the DNase I hypersensitive site (DHS) was found to be 55 bp, and enrichment strongly indicated the potential biological meaning of DHS in 261 diseases and traits [4].

SNP-based methods have been developed to estimate heritability enrichment using measured genotypes of SNPs. This method is more likely to reflect the proportion of phenotypic variation due to causal variants (CVs) tagged by SNPs for nominally unrelated samples. The unbiased estimation of heritability enrichment relies on accurate modeling of genetic architectures. Due to heterogeneity in natural selection, artificial selection and recombination rates across genome regions, heritability enrichment generally varies with minor allele frequency (MAF) and linkage disequilibrium (LD) of SNPs [5]. Previous studies in human genetics found that the relationship (represented by α) between MAF and effect sizes of CVs and the LD heterogeneity in various genome regions affected the unbiased estimation of heritability enrichment of a category [6]. However, there are great differences in population genetic architectures, LD and selection between livestock and humans [7]. The effect of these two factors (i.e., MAF and LD) on heritability enrichment estimation of functional genomic regions of complex traits have not yet been examined on livestock, leading to confusion when estimating the heritability enrichment.

The variance of the allelic effect is proportional to ${{(p}_{i}\left(1-p_{i}\right))}^{1+\alpha }$, where $p_{i}$ is the MAF. The value α can be used to indicate the selection direction of a trait, in that a positive value indicates positive selection and a negative value indicates negative selection [8]. The Classic Genome-wide Complex Trait Analysis(GCTA) model assumes that each SNP is expected to contribute equal genetic variance, thus it considers the α as −1 [9]. Animal and plant genome prediction usually assume α is 0, that is, there is no correlation between MAF and effect size of marker [10]. Speed et al. [5] examined a range of α values and found a negative correlation between MAF and effect size of marker for most traits (mean about −0.25). The α value was not fixed for each phenotype or even different functional regions of the genome [11]. A value of α = 0 or −1 is usually used to estimate the functional enrichment in livestock population [12], but the consequences of different α values on heritability enrichment have not been systematically compared and studied.

Numerous works have also emphasized the importance of considering LD-dependent genetic architectures in the analyses of heritability [13,14]. LD is unevenly distributed across the genome [15]. Classical analysis models for complex traits seldom consider the uneven distribution of LD across the genome [9,10], resulting in overestimation of heritability in high-LD regions and underestimation of heritability in low-LD regions. Recently, some models integrating LD-dependent weights have been applied for the analyses of functional enrichment [16,17]. Thereinto, the LD-adjusted kinships (LDAK) model assumes heritability enrichment varies with the SNP LD, and eliminates the adverse influence of uneven LD distribution by giving a larger weight for SNP in the lower-LD region and a smaller weight for SNP in the higher-LD region [18]. At present, few studies have investigated the performance of the heritability enrichment model considering LD structure in livestock, so it is necessary to investigate how different LD window partitions affect the estimation of heritability enrichment in livestock population.

The quality of sequencing technology and imputation for common SNPs are higher than for uncommon or rare SNPs, so a majority of studies have mainly focused on heritability research for common SNPs. Recently, the effects of uncommon CVs have been studied to gain insight into the genetic architecture of traits and examine genetic networks and annotation categories using imputed SNPs [19,20,21]. With the increase of imputation accuracy of SNP chip data, it is possible to explore the performance of uncommon CVs in heritability enrichment, thus, gaining a more complete understanding of the genetic architecture of complex traits.

Most traits of interest to the livestock industry are complex traits and many CVs are likely to be of small effect and difficult to find [22]. Heritability enrichment is helpful to gain insight into the CVs of complex traits. However, the various assumptions of α value and the existence of an LD-dependent calculation would confuse the result when estimating heritability enrichment in livestock population. We conducted this study with the following aims (1) evaluating the performance of different heritability enrichment models under a variety of scenarios in livestock population and (2) exploring the key factors affecting the performance of heritability enrichment models.

## 2. Materials and Methods

### 2.1. Population and Genotypic Data

The German Holstein population provided by Vereinigte Informationssysteme Tierhaltung Wirtschaftlicher Verein was used in this study [23], which contained 2000 bulls genotyped with the Illumina Bovine SNP50 Beadchip [24]. Taking the 770 k high-density SNP data obtained from the 1000 Bull Genome Project as reference [25], the 54 k SNP data were imputed to 770 k via Beagle software (version 4.0; Browning and Browning; Washington, USA). The mean accuracy of genotype imputation was 0.98 (Appendix A), and the process is explained in detail in [26]. Data quality control were performed by excluding SNPs with MAF < 0.5%, call rates < 90% and significant deviations from Hardy-Weinberg equilibrium (HWE) (*p*-value < 0.000001) using PLINK. Finally, 340,720 SNPs were retained for continued study.

### 2.2. Simulation of Phenotypes

To assess the performance of different models on a range of genetic architectures, we first defined a “Functional subset”, including SNPs mapped from the Kyoto Encyclopedia of Genes and Genomes (KEGG) pathway. Briefly, we downloaded 150 KEGG pathways of dairy cattle using KEGGREST package in R (version 4.0.2; Ihaka and Gentleman; Auckland, New Zealand) and extracted the genes involved in each pathway. According to the physical location of the genes, we mapped 33,544 SNPs located on genic region for these 150 KEGG pathways and defined these as a “Functional subset”. We also set up a “Non-functional subset”, including randomly selected SNPs with the same number of SNPs as in the “Functional subset”.

We then simulated three types of phenotypes, affected by 2000 CVs but having different levels of MAF for CVs (i.e., genome-wide, common, and uncommon variants). Of these 2000 CVs, 500 CVs were selected randomly from the “Functional subset” and 1500 CVs were selected from the rest of the SNPs. The MAF of genome-wide variants ranged from 0.005 to 0.5. We defined “common” as variants with 0.05 ≤ MAF ≤ 0.5, and “uncommon” as variants with 0.01 ≤ MAF < 0.05.

We simulated phenotypes using the following model:y=g+e
where **y** is the vector of phenotypic values; **g** is the vector of true breeding values; **e** is the vector of residual errors with distribution of N (0, $\mathrm{var}\left(g\right)({1/h}^{2}-1$)) for $h^{2}=0.8$. The value **g** could be calculated as $g_{k}=\sum_{1}^{2000}w_{\mathrm{ki}}u_{\mathrm{ki}}$, in which $w_{\mathrm{ki}}{=x}_{\mathrm{ki}}-{2p}_{\mathrm{ki}}$ with $x_{\mathrm{ki}}$ coded as 0, 1, or 2 of individual k at the i-th SNP and $p_{\mathrm{ki}}$ being the frequency of the the i-th SNP; $u_{\mathrm{ki}}$ was the i-th allelic effect size, drawn from $\sqrt{\frac{h^{2}{\times (2p}_{i}\left(1-p_{i}\right){)}^{-1}}{2000}}$. For each CV MAF range scenario, a total of 100 replicated phenotypes were simulated.

### 2.3. Heritability Enrichment Estimation Models

We partitioned the heritability explained by all the SNPs into the “Functional/Non-functional subset” and the rest of the SNPs ($h_{F/\mathrm{Nf}}^{2}$ and $h_{-F/-\mathrm{Nf}}^{2}$). We estimated $h_{F/\mathrm{Nf}}^{2}$ and $h_{-F/-\mathrm{Nf}}^{2}$ by fitting the two different SNP subsets (“Functional/Non-functional subset” and the rest SNP subset) in the genome-based restricted maximum likelihood (GREML) model via LDAK software (version 5.1; Speed; London, UK):$${y=X\beta +Z}_{F/\mathrm{Nf}}g_{F/\mathrm{Nf}}{+Z}_{-F/-\mathrm{Nf}}g_{-F/-\mathrm{Nf}}+e$$
where **y** was the vector of phenotypic values, **β** was a vector of fixed effects (the first ten principal components) with its corresponding coefficient matrix **X**, $g_{F/\mathrm{Nf}}$ and $g_{-F/-\mathrm{Nf}}$ were respectively the vector of additive genetic effect of the selected SNPs and of the rest SNP subset, $Z_{F/\mathrm{Nf}}\;$ and $Z_{-F/-\mathrm{Nf}}$ were, respectively, the design matrices that allocated observations to genetic values, and **e** was the vector of residual errors. The additive genetic and residual values were assumed to be independent normally distributed values: **g**~N (0, $\sigma_{g}^{2}G$) and **e**~N (0, $\sigma_{e}^{2}I$), where $\sigma_{g}^{2}$ and $\sigma_{e}^{2}$ were the additive genetic variance and residual variance, respectively. **G** was a genomic relationship matrix (GRM) constructed from SNP genotypes. **G** was changed in different heritability enrichment models, as detailed below.

There were four models used for estimating heritability enrichment: the VanRaden model, the GCTA model, the α-model and LDAK model. The differences among these four models were the variance of SNP effect sizes and the existence of LD weight calculation. The four models are summarized in Appendix A.

#### 2.3.1. VanRaden Model

The VanRaden model was first proposed by VanRaden, and set the α value as zero, indicating that the variance of the allelic effect (represented as var($\beta_{i}$)) was in proportion to $p_{i}\left(1-p_{i}\right)$. The GRM is constructed as follows:$$G=\frac{{\mathrm{MM}}^{T}}{2\sum_{1}^{m}p_{i}(1-p_{i})}$$
where **M** is the centered genotype matrix, m is the number of SNPs and $p_{i}$ is MAF of SNP_i_.

#### 2.3.2. GCTA Model

For the GCTA model, the standardized genetic variance of SNP_i_ is constant: var($\beta_{i}$)∝1. The GRM is calculated as:$$G=\frac{{\mathrm{MM}}^{T}}{m}$$

#### 2.3.3. The α-Model

The α-model uses an α parameter to model MAF-dependent architectures, which could be considered as: $\mathrm{var}\left(\beta_{i}\right)\propto {{(p}_{i}\left(1-p_{i}\right))}^{1+\alpha }$. According to [5], we also set seven values of α (−1.25, −1, −0.75, −0.5, −0.25, 0 and 0.25), exploring which led to the best model fit. Negative α value meant negative selection, positive α value meant positive selection, and 0 meant no selection was made for the traits. Therefore, we explored the performance of the α-model representing different selection directions and selection intensities. The GRM can be written as:$$G=\frac{{\mathrm{MM}}^{T}}{\sum_{1}^{m}{{[2p}_{i}(1-p_{i})]}^{1+\alpha }}$$

#### 2.3.4. LDAK Model

Speed et al. introduced a new model that considers both LD-dependent and MAF-dependent architectures, named the “LDAK model” [5]. Genetic variance could be written as $\mathrm{var}\left(\beta_{i}\right)\propto {{(p}_{i}\left(1-p_{i}\right))}^{1+\alpha }w_{i}^{*}$, where $w_{i}^{*}$ denotes LD weights reflecting smaller per-SNP heritability for high-LD SNPs. The weight $w_{i}^{*}$ is fixed, based on LD patterns, but not estimated using trait data. The GRM of this model is:$$G=\frac{{\mathrm{MWM}}^{T}}{\sum_{1}^{m}{{[2p}_{i}(1-p_{i})]}^{0.75}}$$
where **W** is the diagonal matrix with elements of $w_{i}^{*}$, $w_{i}^{*}=w_{i}\frac{m’}{\sum_{i}w_{i}}$ and $w_{i}=1/\overset{m’}{\underset{1}{\sum }}{c(i,i’)}^{2}$ with ${c(i,i’)}^{2}$ represents the correlation squared between variant i and i’ and m’ means the SNP number in each LD window.

### 2.4. Model Assessment

The heritability enrichment of the “Functional/Non-functional subset” is defined as the proportion of the explained SNP heritability divided by the proportion of SNPs [27], which could be written as:$$\mathrm{Enrichment}=\frac{h_{\mathrm{subset}}^{2}}{h^{2}}/\frac{N_{\mathrm{subset}}}{N_{\mathrm{total}}}$$
where $h_{\mathrm{subset}}^{2}$ was the heritability of the “Functional/Non-functional subset”, $h^{2}$ was the heritability of all the SNPs, $N_{\mathrm{subset}}$ was the SNP number of the “Functional/Non-functional subset”, which was 33,544; $N_{\mathrm{total}}$ was the SNP number of the total SNPs, which was 340,720.

The true heritability enrichment of three CV MAF ranges scenarios could be calculated, based on the known real effect size and MAF of markers. The true subset heritability was: $h_{\mathrm{subset}}^{2}=\frac{\sum_{1}^{i}{\mathrm{var}(\beta }_{i})}{\mathrm{var}(\mathrm{pheno})}$. The value $\sum_{1}^{i}\mathrm{var}\left(\beta_{i}\right)$ was the total genetic variance of i CVs in “Functional/Non-functional subset”, $\mathrm{var}\left(\mathrm{pheno}\right)$ was the total phenotypic variance. The true total SNP heritability was 0.8.

For easy comparison, we corrected all the true heritability enrichments of the “Functional subset” to 3.000 and of the “Non-functional subset” to 1.000. The study assessed the performance of models by comparing the differences between true enrichment and estimated enrichment.

## 3. Results

### 3.1. Genetic Architectures for Simulated Traits

Using real genotypic data, this study simulated phenotypes with different genetic architectures to explore the performance of heritability enrichment in livestock population. The simulated phenotypes were controlled by 2000 CVs, with the heritability of 0.8. By varying simulation parameters, we obtained three types of phenotypes with different MAF CVs. Table 1 shows the attributes for the simulated phenotypes estimated from the whole data set. T1 phenotype was controlled by markers with MAF ranging from 0.05 to 0.5. T2 and T3 were controlled by common (MAF range: (0.01, 0.05)) and uncommon (MAF range: (0.005, 0.01)) markers, respectively. All phenotypes were determined by 0.59% (*n* = 2000) of markers.

### 3.2. Comparison of Heritability Enrichment Models

We first compared the heritability enrichment results of four models with the default parameters. For comparison purposes, we used the default parameters in the LDAK model (i.e., α = −0.25, LD-window = 100 kb) and α value of −0.25 for the α-model. When the estimated models applied “Functional subset” as the component, there was little difference between all the four models for T1 and T2 phenotypes, and only resulted in a difference of 0.015 to 0.067 from the true values (i.e., 3.000). In particular, the α-model generated the most accurate result. However, for the T3 phenotype, all the four models, except for the GCTA model, generated values that were almost two-thirds of the real results. The GCTA model, combining “Functional subset”, performed the best, overestimating the 3.000 by 0.471 (Figure 1A). The subset with known causal genes, i.e., “Functional subset”, gained more accurate enrichment values than “Non-functional subset”. The enrichment values of models which used “Non-functional subset” as a component were generally unstable, even overestimating by more than six times (Figure 1B). In addition, Appendix A shows the heritability estimation for “Functional subset” and the whole SNP. The LDAK model, with 5000 kb window, performed the best for T1 phenotype. For the T3 phenotype, with the LD windows increased, the $h_{F}^{2}$ fell, while the $h^{2}$ increased (Appendix A).

### 3.3. Effect of α Value Assumption on the Heritability Enrichment Estimation

For different traits, optimal α may differ, we therefore explored the effect of different α value assumptions on heritability enrichment estimation, which was defined as α-model. For ease of comparison, we did not consider the LD weights. For T1 and T2 phenotypes. The model using “Functional subset” with different α values showed no significant difference, which was consistent with previous research [8]. Moreover, models estimating the enrichment of T2 phenotype showed higher accuracy, only 0.006–0.089 off the true enrichment, while estimating the enrichment of T1 phenotype showed a slight downward trend with an increase of α values. However, for the phenotypes controlled by the uncommon CVs and estimation using the “Functional subset”, with the increase of α values, the estimated enrichment values decreased from 3.467 to 1.541 (Figure 2A), while using “Non-functional subset” showed the opposite trend. When α = −0.75, the model incorporating “Functional subset” showed the most accurate heritability enrichment value of 3.202. Using the “Non-functional subset” as a component, it was observed that different α values influenced and ruled heritability enrichment. Compared with “Functional subset”, their estimates showed great bias (Figure 2B).

### 3.4. Effect of LD on the Heritability Enrichment Estimation

We then investigated the potential ramifications of LD-dependent architectures on heritability enrichment. As Appendix A shows, LD decayed with distance. When the SNP distance enlarged to about 150 kb, the r^2^ between SNPs decayed to 0.2. LD decay distance was approximately 40 kb in this population, with the criterion of 0.5 r^2^. When establishing the GRM, we set six different LD windows (1 kb, 5 kb, 50 kb, 100 kb, 500 kb and 5000 kb) which indicated different SNPs weights. For example, setting 1 kb window meant that the genome was divided into some 1 kb windows, and the software would accord the average r^2^ of all SNP pairs to weight each SNP for each window. SNP weight tended to be higher for the SNP in a region of low LD, and, thus, the model assumed that this SNP contributed more than those in high-LD region. In this way, the model assigned a different weight to each SNP and excluded SNPs with a weight of zero. In addition, we applied the default α parameter (−0.25) of the LDAK model to explore the effect of LD.

For T2 phenotype, enrichment values were widely overestimated by the LDAK model. As the LD window increased, the enrichment of the model using “Functional subset” slowly raised to 3.926. What was more, the results of the T1 phenotype showed the same trend. As for the T3 phenotype, with the LD windows enlarged, the estimated enrichment reduced (Figure 3). Compared to the α-model, which assumed the α = −0.25, considering the effect of LD did not increase the accuracy of enrichment estimation except for uncommon CVs. For the T3 phenotype, models considering the LD-dependent genetic architecture acquired more accurate estimation (Figure 4).

## 4. Discussion

As more and more heritability enrichment models are proposed, there is a need to assess the performance of different models for the livestock industry. In this study, we used simulated phenotypes with a variety of underlying genetic architectures to evaluate four heritability enrichment models with different parameters. According to the results of this study, the estimates of heritability enrichment showed bias and depended on the model and trait genetic architecture.

In this study, we explored how different models and trait genetic architectures effected the estimates. Previous livestock studies often used VanRaden and GCTA models for heritability analysis [28,29]. This study showed that different models with different parameters would influence the performance of heritability enrichment. Otherwise, different trait genetic architectures also led to different enrichments. For T1 and T2 phenotypes, the estimated enrichments were more precise and robust, while for the T3 phenotypes, the enrichments were always underestimated. For the T1 phenotype, the α-model, with α = −1.25, performed the best, being only 0.002 off 3.000. The α-model with α = −0.5 produced the most accurate result for the T2 phenotype. For the T3 phenotype, considering the LD weights could get more accurate enrichment data. Therefore, we assumed that by considering both the α value and LD weight the uncommon CVs would obtain the best result. When SNP subset explained a relatively large amount of genetic variance (greater than the mean value), it could be seen that the SNP subset was the region associated with the corresponding phenotype. Therefore, the overestimation or underestimation of heritability would lead to the bias of the estimated enrichment value. Additional factors that we did not investigate might also influence heritability enrichment estimation, such as type of SNP data, degree of sample stratification, shared environmental effects, technical artifacts, environmental factors that co-vary with ancestry, CVs with MAF < 0.005, or non-SNP CVs [30,31,32].

Varying the value of α in the α-model would change the assumed relationship between the MAF and effect size of CVs, and, then, the heritability estimation and the heritability enrichment value would be affected. The value of α is applied to measure selection [8,11] for which larger α indicates that common SNPs tend to have larger effect sizes than less common SNPs, and vice versa. Therefore, uncommon CVs with positive α would obtain smaller effect sizes, resulting in lower heritability enrichment (Figure 2). The GCTA and VanRaden models are the widely used models in the livestock industry. The GCTA model assumes the effect size of each SNP to be consistent, and the VanRaden model is a neutral model, assuming that no selection is made for the traits. Zhang et al. showed the existence of negative selection in the dairy cattle population [33], so we took α = −1 to simulate the phenotype which meant negative selection in the population and equal variance of breeding value for 2000 causal variants. Our result showed the enrichment was more accurate when the simulation parameters were consistent with the calculated model parameters. Indeed, α value is not fixed for different phenotypes, being affected by many factors (such as selection direction and strength and so on). Ideally, the α-model should be employed for each trait; however, introducing a multiparameter heritability model would require substantial algorithmic changes and would dramatically increase computational demands. With this in mind, we recommend using the GCTA or VanRaden model for the T1 and T2 phenotypes. Nevertheless, for the T3 phenotype, too high or too low an α value leads to the underestimation or overestimation of CV effect sizes. Therefore, we recommend testing the α parameter to obtain more accurate heritability enrichment.

Unlike humans, livestock have mostly undergone natural and artificial selection. Hence, there is extensive positive and negative selection in genetic variant associated with complex traits, so the α value of a trait (i.e., the direction of selection) is the result of the many signals of positive and negative selection on trait-associated variants cancelling each other out [34]. In addition, the estimation of the α value may also be affected by the genetic correlation and selection strength among traits, as well as the SNP data type [35,36].

Common CVs are widely distributed in the genome that are greatly affected by LD. As the LD window size increases, the LD between SNPs in each window weakens. Therefore, the LDAK model allocates higher weights to CVs among the whole genome, so then the heritability enrichment increases. Enlarging the LD window strengthens the LD of uncommon CVs and low-frequency SNPs, thus decreasing the heritability enrichment of uncommon CVs (Figure 3). Nevertheless, the estimations of heritability enrichment of uncommon CVs are often underestimated, so considering the effect of LD would reduce the bias (Figure 4). Here we only considered the LDAK model to explore the effect of LD on heritability enrichment. Other models with different LD structure assumptions from the LDAK model, such as baseline-LD model [21], stratified LD-score regression [37], GCTA-LDMS [16] and LD score regression (LDSC) model [38], are also worth exploring in the future.

We used raw genotype files and chip SNP data to explore the performance of different heritability enrichment models. In reality, as more and more Genome-wide association analysis (GWAS) data from public databases are used, heritability enrichment models using summary statistics estimation have come into being [11,27,36]. Therefore, there is a need to explore the performance of heritability enrichment models using summary statistics for livestock in the future. Recently, Hou et al. [39] proposed GRE, a method for estimating SNP heritability without specifying a heritability model, and the performance of heritability enrichment of this method is also worth exploring. What is more, with the increasing availability of whole-genome sequence (WGS) SNPs, more genetic effects are likely to be captured and provide the most accurate heritability enrichment, especially for rare CVs.

## 5. Conclusions

In this study, we investigated the performance of different heritability enrichment models, with a series of parameters, on three different phenotypes. Models with a subset of known causal genes acquired the more accurate and stable enrichment values compared with using the “Non-functional subset”. Heritability enrichments were sensitive for the α value assumption and LD weight. This is a prospective study on livestock, which can provide a reference for heritability enrichment analysis of complex traits of livestock.

## Figures and Tables

**Figure 1 genes-13-01644-f001:**
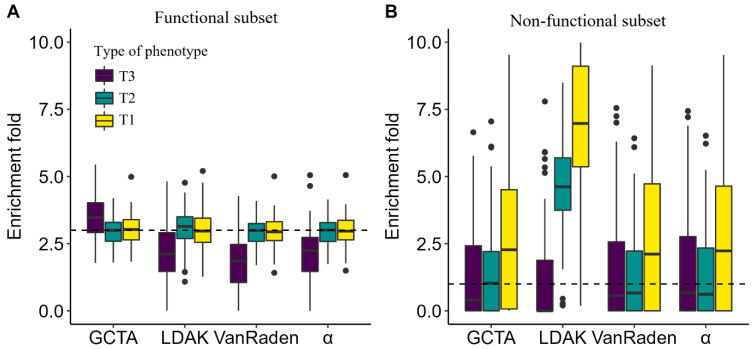
Comparison of heritability enrichment estimation models. Showing the estimated enrichment for different models integrated with (**A**) “Functional subset” or (**B**) “Non-functional subset”. Colors represent the type of phenotype. The dotted lines represent the true enrichments (i.e., 3.000 and 1.000, respectively).

**Figure 2 genes-13-01644-f002:**
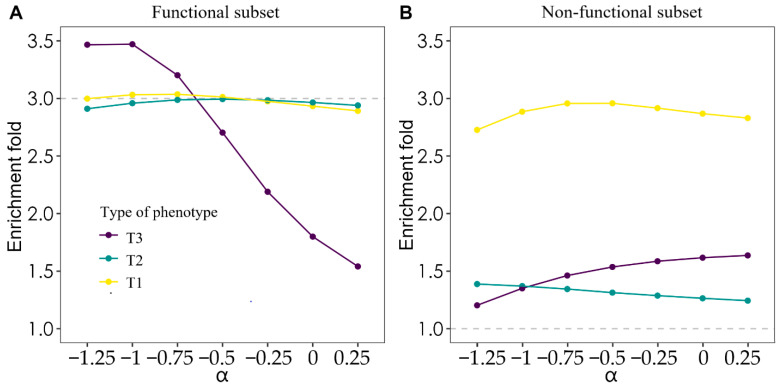
Comparison of performance of different α values for α-model. Showing the estimated enrichments of different α values for α-models integrated with (**A**) “Functional subset” or (**B**) “Non-functional subset”. The different color and dotted lines are explained as in Figure 1.

**Figure 3 genes-13-01644-f003:**
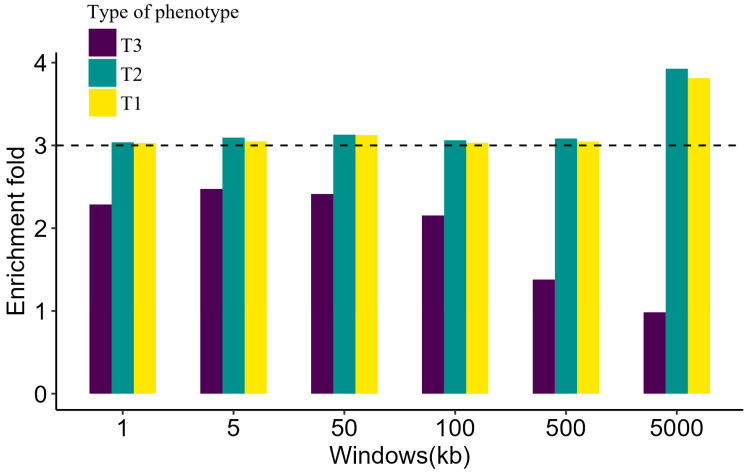
Comparison of different LD window settings for the LDAK model. The estimated enrichment of different LD window settings for the LDAK model integrated with “Functional subset”. The different colors and dotted line are explained as in Figure 1.

**Figure 4 genes-13-01644-f004:**
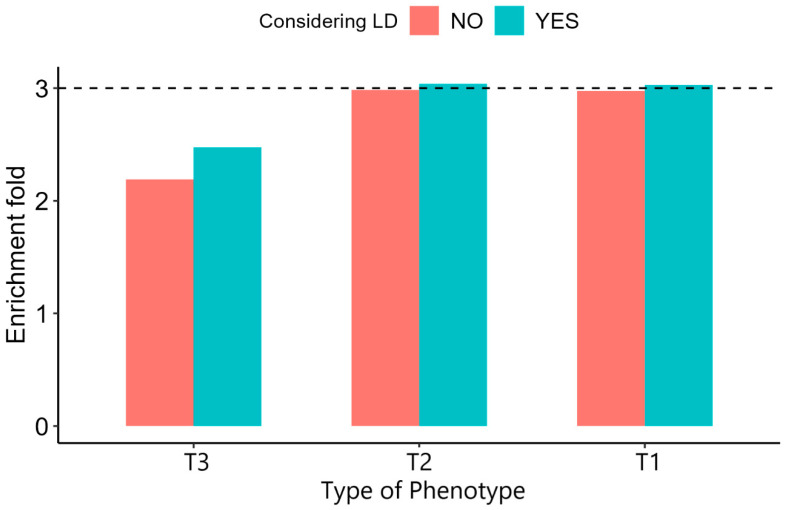
Performance of enrichment model with (LDAK) and without (α-model) LD weights. Comparing the difference between the real enrichment and the estimated enrichment of the model with and without LD weights. Colors represent the existence of LD weight for models. The dotted line represents the true enrichment.

**Table 1 genes-13-01644-t001:** The attributes for each type of simulated phenotype.

Type of Phenotype	MAF of Causal Variants ^a^	Number of Causal Variants ^b^	True Enrichment Fold ^c^	
			Functional Subset	Non-Functional Subset
T1 (genome-wide)	(0.005, 0.5)	2000	3.000	1.000
T2 (common)	(0.05, 0.5)		3.000	1.000
T3 (uncommon)	(0.01, 0.05)		3.000	1.000

^a^ The minor allele frequency range of causal variants; ^b^ The number of markers that control the phenotype; ^c^ True enrichment values for different subsets.

## Data Availability

Publicly available datasets were analyzed in this study. The SNP chip data of 2000 bulls can be found at: https://www.g3journal.org/content/suppl/2015/02/09/g3.114.016261.DC1 (accessed on 5 February 2015). The KEGGREST code used in this study is available at: https://github.com/SCAU-AnimalGenetics/Heritability-enrichment (accessed on 29 August 2022).

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
