# Peer review of "Model Comparison of Heritability Enrichment Analysis in Livestock Population"

_genes, 2022, doi:10.3390/genes13091644_

Round 1
Reviewer 1 Report
Please explain the detail of selection for the studied German Holstein population, which seems important to create LD and haplotype construction over the genome. Also, I propose to evaluate the degree of LD in this population. As also the authors denoted, the livestock population is undergoing selection, which could create LD patterns even across chromosomes (Bulmer effect).
Did the authors perform Hardy-Weinberg equilibrium tests?
Why did the authors set the heritability to be 0.8?
In 2.3.1, M was centered and "scaled", was it true? I have recognized that in Vanraden's method 1, the genotype did not scaled, but not for Vanraden's
method 2.
In 2.3.2, was dividing by 2 correct? Please check this.
The authors should show not only the enrich level but also the estimated values themselves.
Please discuss the impact of how to generate SNP effects in simulating phenotype. In this study, variance of breeding value of each SNP was totally the same under Hardy-Weinberg equilibrium condition.
Author Response
Dear Reviewer,
The point-by-point response to the your comments was uploaded as a Word file.
We hope you find the revised manuscript acceptable for publication in Genes. We are looking forward to your further comments.
With kind regards, on behalf of the authors,
Zhe Zhang

Reviewer 2 Report
The authors aimed to explore the performance of different heritability enrichment models in livestock population, we used the VanRaden, GCTA and α models assumed different α values and LDAK model considering LD weight. As a result they found heritability enrichment estimation were sensitive for the α value assumption and LD weight consideration of different models, where accuracy would be greatly improved by using suitable model.
The Introduction section is sufficient.
The Material and Methods section is well described.
1. Line 70: “Speed et al. examine” should be “Speed et al. [5] examine”
2. Line 120: Could you provide KEGGREST codes in Supplementary file.
3. Line 141: h2Functional/Non-functional may be written as h2F/Nf for easy written and shortage the equations.
4. Line 242: Please give some information in method section about the selection of seven values of α (−1.25, −1, −0.75, −0.5, −0.25, 0 and 0.25) and why that values?
5. In Figure 2; the caption of X axis should be a instead of a.
Author Response

(The authors gave the same response as above.)

Round 2
Reviewer 1 Report
The manuscript has been improved.
Spell-checking seems to be required throughout the manuscript.